# SnO_2_-Based NO_2_ Gas Sensor with Outstanding Sensing Performance at Room Temperature

**DOI:** 10.3390/mi14040728

**Published:** 2023-03-25

**Authors:** Rahul Kumar, Raman Kumari, Vidya Nand Singh

**Affiliations:** 1CSIR-National Physical Laboratory, Dr. KS Krishnan Marg, New Delhi 110012, India; 2Academy of Scientific and Innovative Research (AcSIR), Ghaziabad 201002, India

**Keywords:** metal oxide, nanomaterial, NO_2_ gas sensor, spin-coating

## Abstract

The controlled and efficient formation of oxygen vacancies on the surface of metal oxide semiconductors is required for their use in gas sensors. This work addresses the gas-sensing behaviour of tin oxide (SnO_2_) nanoparticles for nitrogen oxide (NO_2_), NH_3_, CO, and H_2_S detection at various temperatures. Synthesis of SnO_2_ powder and deposition of SnO_2_ film is conducted using sol-gel and spin-coating methods, respectively, as these methods are cost-effective and easy to handle. The structural, morphological, and optoelectrical properties of nanocrystalline SnO_2_ films were studied using XRD, SEM, and UV-visible characterizations. The gas sensitivity of the film was tested by a two-probe resistivity measurement device, showing a better response for the NO_2_ and outstanding low-concentration detection capacity (down to 0.5 ppm). The anomalous relationship between specific surface area and gas-sensing performance indicates the SnO_2_ surface’s higher oxygen vacancies. The sensor depicts a high sensitivity at 2 ppm for NO_2_ with response and recovery times of 184 s and 432 s, respectively, at room temperature. The result demonstrates that oxygen vacancies can significantly improve the gas-sensing capability of metal oxide semiconductors.

## 1. Introduction

Due to the increasing population, industrialisation, vehicles, air conditioners, and other sources, many gases are emitted, which affects the environment. Research is being carried out to detect and mitigate these gases. Contributions from fossil fuel combustion, automotive emissions (internal combustion), explosions, welding, and other human-caused sources are also significant. The requirement for monitoring/controlling the ambient environment (temperature, atmospheric pressure, and humidity) in museums, organic farming, the paper industry, sophisticated instruments, pharmaceuticals, electronics manufacturing, packaging, research laboratories, the medical industry, and standard/calibration laboratories have also led to the development of novel and advanced sensing techniques [1,2,3]. Thus, measuring the atmosphere’s toxic gases is necessary to save lives and nature. Different types of toxic gases, such as CO, H_2_S, NH_3_, NO, SO_2_, and NO_2_, are in the atmosphere. Nitrogen dioxide (NO_2_) is a crucial air pollutant among these. A gas sensor can easily detect these harmful gases. The device that detects the output signal electronically is called a gas sensor. The concentration of NO_2_ in ambient air should not exceed 40 g/m^3^ annually and 80 g/m^3^ on average in 24 h for residential/industrial areas (according to the National Ambient Air Quality Standards (NAAQS), 2009 notified by the Central Pollution Control Board (CPCB), India). According to the US EPA (Environmental Protection Agency), NO_2_ levels in ambient air should not exceed 53 parts per billion (ppb) yearly and 100 parts per billion (ppb) hourly. Thus, the exceeded limit of this toxic gas is harmful to us. Typically, semiconducting material-based devices are used to detect these gases. Some systems are equipped only with one sensor, which makes the instruments vulnerable to changing humidity backgrounds and interfering gases. These can cause a false alarm.

As a result, scientists are working to create an effective NO_2_ gas sensor employing several metal oxides (MOs). More than 5200 research publications on the MO-based NO_2_ gas sensor have been published to date (Jan. 2022) [Source: Scopus data]. China is a world leader in using MOs to detect NO_2_ gas. According to Scopus data, Chinese authors produced over 1300 research articles on MOs-based NO_2_ sensors, followed by Korean scientists with over 600 articles. Meanwhile, scientists in the United States and India produced around 500 studies on the same NO_2_ gas sensor [4]. Semiconductor materials are highly researched among gas sensors due to their robust nature and good performance. For gas-sensing applications, semiconductor material structure and surface modification are essential. Several innovative techniques based on the optical and optoelectronic properties of these materials are gaining wide acceptance [5,6] because of their high surface-to-volume ratio. Sensors based on nanostructured materials have a higher response than bulk materials due to highly active large surface areas and selective sensitivity to certain gases [6,7]. Metal oxide nanomaterials with various morphologies, such as nanoparticles [8], nanorods [9], nanowires [10], nanoflowers [11,12], and nanotubes [13], have demonstrated outstanding sensing characteristics because of their excellent optical and electrical capabilities [14].

Tin dioxide (SnO_2_) is a typical n-type nanomaterial with a large bandgap (3.6 eV) and inherent oxygen vacancies [15]. SnO_2_ thin films cover a wide range of optoelectronic device applications, such as gas sensors, solar cells, heat-reflective mirrors, liquid crystal displays (LCD), light detectors, transparent conducting electrodes, and far-infrared detectors [16,17]. High-efficiency solar cells and biosensors made of SnO_2_-based films have sparked researchers’ interest. SnO_2_ thin film is a viable candidate for gas detection, since SnO_2_ is a semiconductor with a high capacity for oxygen absorption [18]. As the surface–gas interaction is poor in some materials, primarily absorption along with minor chemisorption occurs, resulting in electronic instability and making detection nearly impossible. As a result, developing a gas sensor that is both sensitive and effective is required to identify the gas [19]. Due to its excellent structure and optoelectronic capabilities, SnO_2_ thin film is particularly effective. Some other characteristics of SnO_2_ include high visible transmittance, high infrared reflectivity, high mechanical hardness, low electrical resistivity, wide bandgap, and high chemical, thermal, and good environmental stability.

The device in this work is fabricated by spin-coating a nano-powder dispersion. For the device, we used a comb-type aluminium metal contact. The sample’s optical, structural, and microstructural properties were determined to obtain more information about its morphology, purity, and other parameters. After that, gas-sensing measurements and their selectivity and sensitivity with different responses and recovery rates were performed. The novelty of this study is that an improved SnO_2_-based NO_2_ gas sensor was developed at room temperature using a simple technique and low-cost base material.

## 2. Experimental Work

### 2.1. Synthesis of SnO_2_ Nano-Powders and Fabrication of Sensing Device

Tin oxide nanomaterial was synthesised by the sol-gel method. In this synthesis method, first, 100 mL of distilled water (18 Ω) was used to fill 500 mL round-bottom flasks. Then 2 g (0.1 M) of stannous chloride dehydrated (SnCl_2_·2H_2_O) was dissolved at room temperature with constant magnetic stirring. After complete dissolution, ammonia solution was added dropwise with continuous magnetic stirring. Some precipitate was formed and centrifuged. The resulting gels (precipitate) were dried at 80 °C for 35 h to remove moisture (water molecules) and some impurities. After that resultant precipitate was heated to 500 °C for 3 h in a box furnace to obtain fine SnO_2_ nanoparticles. Structural characterisation of SnO_2_ was performed using the X-ray diffraction technique. Microstructure and surface morphology was analysed using scanning electron microscopy (SEM). The transmission electron microscopy technique (TEM) revealed microscopic crystallographic information.

For the deposition of SnO_2_ film, a 1 × 1 cm^2^ glass wafer was rinsed with the soap solution and then DI water. After that, the substrate was ultrasonicated in acetone, IPA, and DI water for 10 min each and was dried using Nitrogen gas. After that, the dispersed tin oxide in ethanol was spin-coated, and the sample was kept in a box furnace for annealing at nearly 500 °C for 2 h. Then, the aluminium electrode was thermally deposited using a comb-like mask with a finger gap of 0.5 mm.

### 2.2. Characterisation Techniques

X-ray diffraction (XRD) measurements of synthesised nano-powder and SnO_2_ thin films were performed utilising the Rigaku Ultima IV with Cu-Kα radiation (λ = 1.5406 Å) to study the crystalline phase purity. The samples were scanned at 40 mV/40 mA with a step size of 0.02° in the 20° to 80° range. The Williamson–Hall (W–H) method was used to calculate crystallite size and strain in the samples. SEM (Zeiss EVO MA 10) was used to examine the sample’s morphology. To evaluate nanoparticle production and microstructural characteristics, TEM (FEI, TF 30, s-twin) was used to obtain low- and high-resolution TEM images. Using EDS (Oxford INCA 250) attached with SEM (Zeiss), the EDS spectra elemental information of the as-prepared sample was obtained. Renishaw in Via Raman Reflex Raman spectrometer was used for Raman spectroscopy in the 200–900 cm^−1^ range for structural properties. Using a static gas sensor measurement system, the sensing characteristics of samples were measured in a closed chamber at room temperature. The static gas-sensing system provided by Ants Innovation Private Limited was used. A Keithley 2450 source meter was used to record the change in resistance by applying a constant voltage of 0.5 V. With the controlled gas-injection procedure, NO_2_ concentration was varied. The relation shown in Equation (1) was used to calculate the sensors’ sensitivity:(1)S=(Rg−Ra)×100/Ra

R*a* and R*g* denote the sensor’s resistances in air and gas, respectively. All gas-sensing device measurements were performed at room temperature, and gas-sensing properties were tested very carefully with all precautions in a standard-size airtight stainless steel chamber. NO_2_ was used as the testing gas, and different quantities of NO_2_ were inserted into the test chamber using a microsyringe. The system includes a chamber, a sensor element, and a source meter. SnO_2_-based sensing elements were precisely positioned so that the gas passed through the maximal surface area of the sensing elements.

The sensor’s time to reach 90% of its maximum resistance value while exposed to the target oxidising gas was used to measure the response time. Upon achievement of the maximum resistance value, the target gas was evacuated from the test chamber, and the sensor was allowed to return to its starting resistance value in atmospheric air while maintaining the temperature. In atmospheric air, the sensor’s recovery time is defined as the time it takes to reacquire 10% of the initial resistance value.

## 3. Results and Discussion

### 3.1. Phase Identification by XRD and HRTEM

The synthesised nanomaterial’s phase purity and crystalline nature were probed using powder XRD. This confirmed that the SnO_2_ structure is tetragonal. These agree with the standard JCPDS file No. 41-1445. Figure 1a depicts the XRD patterns of SnO_2_ powder samples. The strong peaks obtained for the samples show high crystallinity. The Bragg diffraction angle of 2 theta values 26.61°, 33.89°, 37.89°, 38.96°, 51.78°, 54.75°, 57.81°, and 61.87° corresponds to planes (110), (101), (200), (111), (211), (220), (022) and (310) respectively. No extra peaks were identified in the samples, confirming the phase purity of the produced samples. Furthermore, the crystallite size was computed using the Williamson Hall method (Figure 1b), as shown by Equation (2),
(2)$$\beta cos\theta =4\varepsilon sin\theta +\frac{k\lambda }{d}$$
where *β* is full width at half maxima, *θ* is the Bragg angle, *ε* is the strain, *k* is the shape constant (*k* = 1), and *d* is the crystallite size. The calculated crystallite size and strain are 12.36 nm and 0.0020, respectively. The refined XRD pattern for SnO_2_ and the schematic diagrams of a unit cell is shown in Figure 1c and Figure 1d, respectively. Rietveld refinement was performed on the experimental XRD data of the SnO_2_ sample. The fitting between simulated and experimental data was found to be very similar. The schematic diagram of the fabrication of the device is shown in Figure 1e.

Refinement-related parameters were in the reliable range, as given in Table 1. A reference file corresponding to SnO_2_ was generated using Vetsa software, and refinement was performed using Full prof software [20]. Background fitting was done using linear interpolation, and the Pseudo Voigt function was used for peak shape fitting of the XRD data [21]. Table 1 shows the refinement parameter and a few structural-related essential parameters.

The fluctuation in crystallite size implies that the solvent composition significantly affected the tetragonal shape and size of the tin oxide nonmaterial. Furthermore, the generally used Scherer equation does not account for instrumental and strain broadening of the diffraction peak, resulting in incorrect crystallite size estimations. The TEM results are shown in Figure 2a. The corresponding histogram is shown in Figure 2b. The particle size is in the 8–18 nm range, with the most probable size being 12 nm. HRTEM was used to gain additional insights into structural information, as shown in Figure 2c. The marked lattice spacing matches well with the XRD results described in Figure 1. Similarly, the SAED pattern in Figure 2d shows the lattice planes as observed in XRD results.

### 3.2. Microstructural Investigations by Scanning Electron Microscopy

The SEM technique was used to analyse the surface morphology and structural information of the SnO_2_ thin film (Figure 3a–c). Particles are well connected with each other. The surface of the film is not uniform and possesses porosity, which makes it more sensitive towards gases. The SEM micrographs show the connectivity among particles [22]. In Figure 3c, the aluminium mask used is also visible. An EDS plot for SnO_2_ is shown in Figure 3d and consists of peaks corresponding to Sn and O.

### 3.3. Raman Spectra of SnO_2_ Material

In the Raman spectrum, Raman modes have been observed at 480 cm^−1^, corresponding to O-Sn-O stretching along the molecular axis. Other dominant modes with the highest intense peak have been observed at 633 cm^−1^ for the acoustic mode (A_1g_), and one B_1g_ mode has also been observed at 770 cm^−1^, as shown in Figure 4 [23]. All these modes have demonstrated the formation of SnO_2_ as proposed from SEM and TEM measurements [24].

### 3.4. UV Spectroscopy Measurements

The UV measurements were performed to evaluate the electronic bandgap of the SnO_2_ nanoparticles. Absorbance versus wavelength data obtained from the UV measurement was fitted according to the Tauc plot to reveal whether it follows the direct or indirect bandgap. From the fitting curve, as shown in Figure 5, it is clear that the SnO_2_ has a direct bandgap because our data has been best fitted with the square of the product of energy and absorbance versus energy corresponding to the incident wavelength. The slope of the curve intersects the energy axis, and the bandgap value is 3.58 eV. However, several low-intensity peaks have been demonstrated near the range of 3.5–4.0 eV, which might be due to the intraband transition and exciton association due to lower energy absorption in comparison to the bandgap of the base material.

### 3.5. Gas-Sensing Measurement

A static gas-monitoring system of 250 mL stainless steel was used to test gases made by Ants Innovative Private Limited. NO_2_ and other gases were injected through a microsyringe at different ppm into the test chamber and measured by using the formula C_1_V_1_ = C_2_V_2_, where C_1_ and C_2_ are concentration of purged gas and the cylinder concentration respectively, and V_1_ and V_2_ are the volumes of the test chamber and microsyringe, respectively. The device electrodes were connected by copper wire by high-grade silver paste before carrying out the study for monitoring gas-sensing measurement. The measurement was recorded by a KEITHLEY 2400 source meter. The room temperature condition was 21 °C, and humidity was 60%, measured by a fluke 1620A “Dewk” thermo-hygrometer.

SnO_2_ is an n-type semiconductor, and NH_3_, NO, H_2_S, and NO_2_ (oxidising gas) were used in the present work. When the device is exposed to the NO_2_ gas, the resistance increases. The gas-sensing mechanism depends on the adsorption and desorption of NO_2_ gas (oxidising gas) on the SnO_2_ (n-type) film surface. NO_2_ traps the free electrons from the SnO_2_ surface and increases the sensor resistance. At room temperature, O_2_^−^ is present on the SnO_2_ surface, but NO_2_ gas molecules react directly with Sn ionic sites instead of oxygen species. The reaction is
(3)$${NO}_{2}+{Sn}^{2+}\to \left({Sn}^{3+}-{NO}_{2}^{-}\right)\to \left({Sn}^{3+}-O^{-}\right)+NO$$

The reduction in the concentration of charge carriers from the conduction band of SnO_2_ upon adsorption results in an increase in sensor resistance. When gas is released due to desorption, the device gains initial condition at room temperature, called recovery.

Researchers have produced NO_2_ sensors by dropping the dispersion over the electrode to illustrate the sensing application of SnO_2–_rGO nanocomposites. The SnO_2–_rGO_2_ responded to 5 ppm NO_2_ at operating temperatures ranging from 30 °C to 60 °C [25,26]. In such a case, the NO_2_ competes with oxygen (O_2_) in adsorbing onto the surface of SnO_2_, complicating the oxidation kinetics analysis.

The schematic of the device used in this work is shown in Figure 6, and the SEM of the device is shown in Figure 3c. The change in the resistance at room temperature with the exposure of NO_2_ is shown in Figure 7. According to the analysis of adsorption–desorption theory, the sensing performance of metal oxide semiconductors can be increased by increasing the gas adsorption site, creating more oxygen vacancies, and boosting the surface activity to catalyse the reaction. The resistance increases as the concentration of NO_2_ increases, which may be quantified for absolute gas-sensing measurement. The device has almost linear behaviour at room temperature, with a significant change in resistance with a change in gas concentration. The SnO_2_ gas sensor tested NO_2_ gas for 0.5 to 2 ppm at room temperature, as shown in Figure 7.

The fluctuation in resistance of metal oxide thin films, when exposed to NO_2_ gas, makes them ideal for gas sensing. This study describes a novel and simple gas sensor based on SnO_2_ nano-powder thin films that serve as a physisorption/chemisorption-based sensing element. In the relative gas range, the SnO_2_ nano-powder thin-film-sensing element with Al masks demonstrates exceptional linear sensing performance for 0.5 to 2 parts per million (0.5, 1, and 2 ppm). Furthermore, at two ppm at ambient temperature, the most incredible sensitivity of 190 was achieved. The presence of oxygen vacancies and the microporous nature of the film allows gas molecules to be absorbed into the film, causing the gas to react at a pace that varies with the amount of gas present. The experimental results show that the thin layer improves sensing capability in SnO_2_ powder by increasing the surface area of the nanoparticles. SnO_2_-based sensing elements are resistant to ageing and have highly reproducible properties.

Furthermore, the gas sensor has a response time of 184 s and a recovery time of 432 s at room temperature, which is a relatively swift response. The sensor performance of 1 ppm of H_2_S, NO, NO_2_, and NH_3_ at 150 °C and 1 ppm of NO_2_ at 50 °C were carried out, and the results are shown in Figure 8a–e. The high-temperature sensing performance of different gases did not fully recover, as shown in Figure 8. The response time improved compared to room temperature NO_2_ detection, but recovery was not complete. The results of this study and its comparison with other researcher’s results are presented in Table 2.

## 4. Conclusions

Herein, we have demonstrated a high-performance gas sensor that utilises metal oxide. We used SnO_2_ as a gas-sensing element to detect various gases (NH_3_, NO, NO_2,_ and H_2_S) at various temperatures. The change in resistance is quantified, corresponding to the gas concentration change at room temperature. The fabricated device is selectively sensitive to NO_2_ gas. The material used is one of Earth’s most abundant materials. The device can be easily fabricated by spin-coating the dispersed SnO_2_ nano-powder (prepared by sol-gel method) in ethanol on the glass substrate. Its ability to function at room temperature is the device’s primary feature. Overall, from the NO_2_ measurement and detailed materials property characterisation, we have established that the sensing device with a smaller crystallite size and porosity provides better sensing performance. The sensor showed high sensitivity at 2 ppm with response time and recovery times of 184 s and 432 s for NO_2_ gas at room temperature. This work can help in producing a low-cost gas-sensing device.

## Figures and Tables

**Figure 1 micromachines-14-00728-f001:**
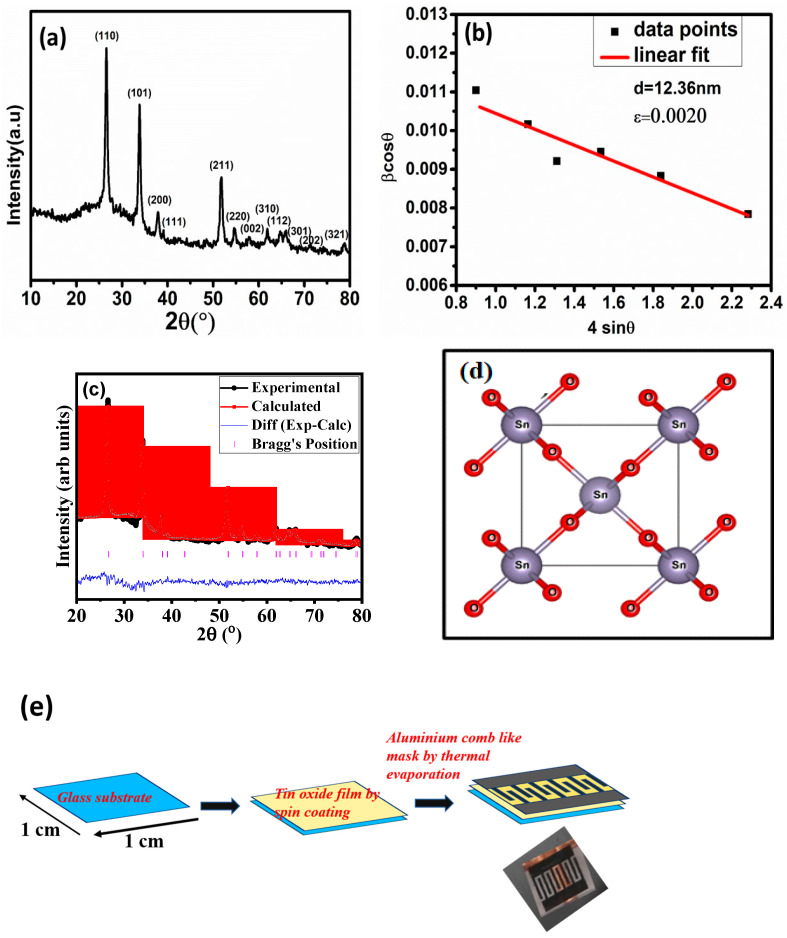
(**a**) XRD patterns of SnO_2_ nano-powder with indexed diffraction peaks corresponding to standard JCPDS data card (41-1445). (**b**) W-H plot of the synthesised material. (**c**) Refinement of SnO_2_ compared with the standard Bragg position. (**d**) Schematic diagram of a unit cell of SnO_2_ originated using Vesta software. (**e**) Schematic diagram of the fabrication of the device.

**Figure 2 micromachines-14-00728-f002:**
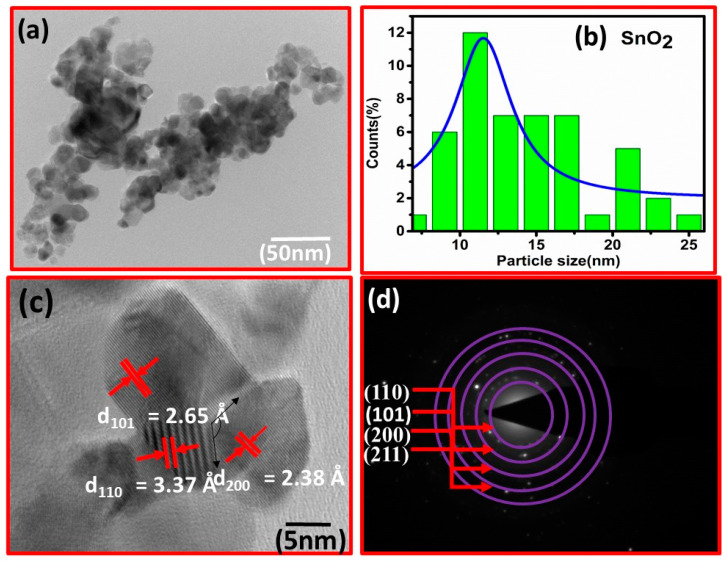
(**a**) TEM image of sensing base material. (**b**) Histogram analysed by Image-J software. (**c**) HRTEM image of nanoparticles. (**d**) Shows a different plane in the SAED pattern.

**Figure 3 micromachines-14-00728-f003:**
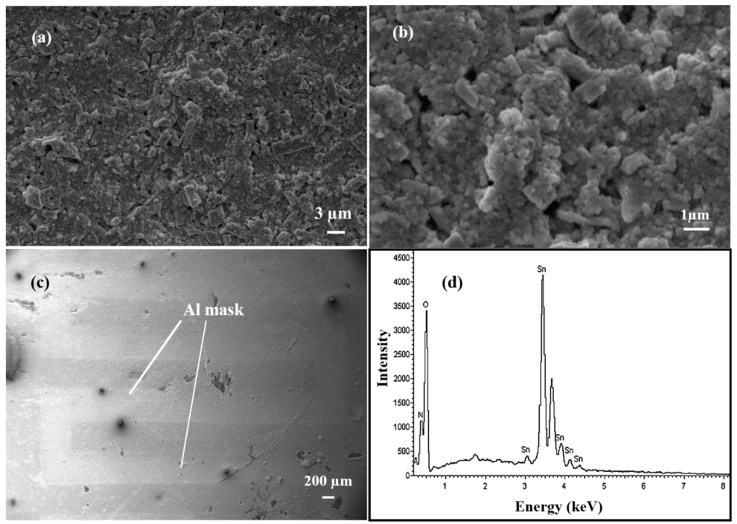
(**a**–**c**) 6 SEM image of the SnO_2_ thin film at different magnifications. (**d**) EDS of the tin oxide film.

**Figure 4 micromachines-14-00728-f004:**
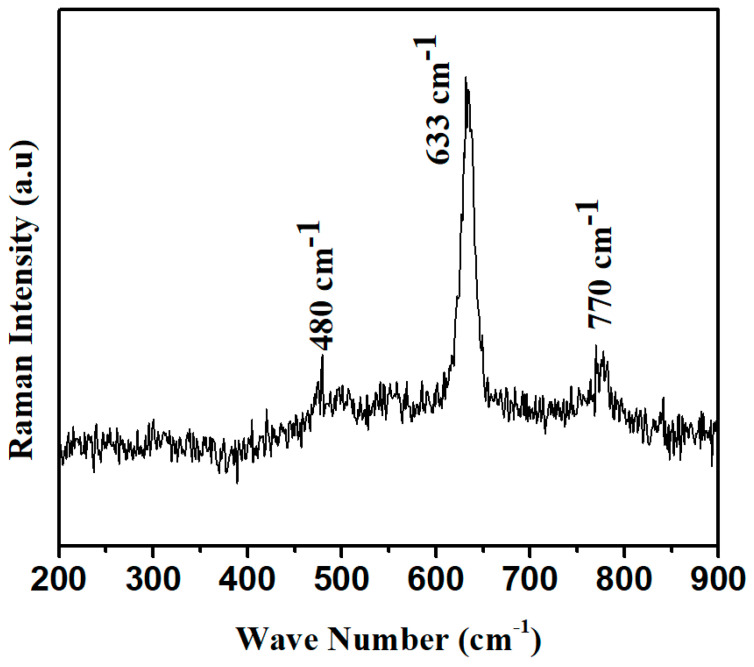
Raman spectra of SnO_2_ nano-powder.

**Figure 5 micromachines-14-00728-f005:**
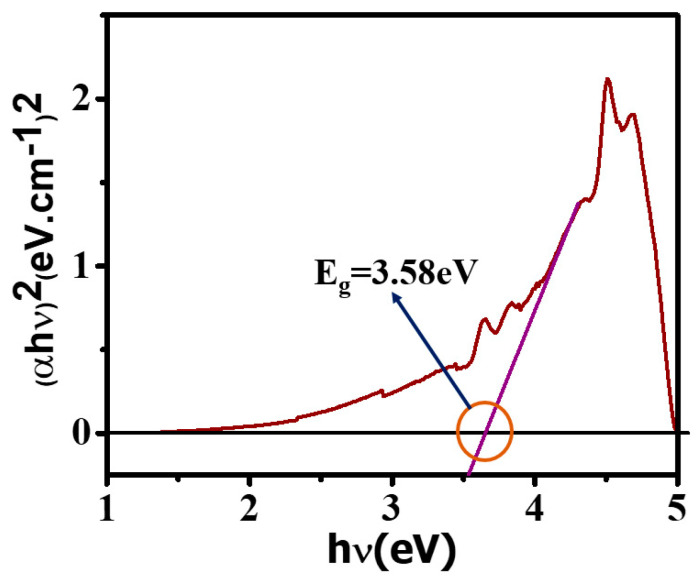
UV spectroscopy for the confirmation of the SnO_2_ band gap.

**Figure 6 micromachines-14-00728-f006:**
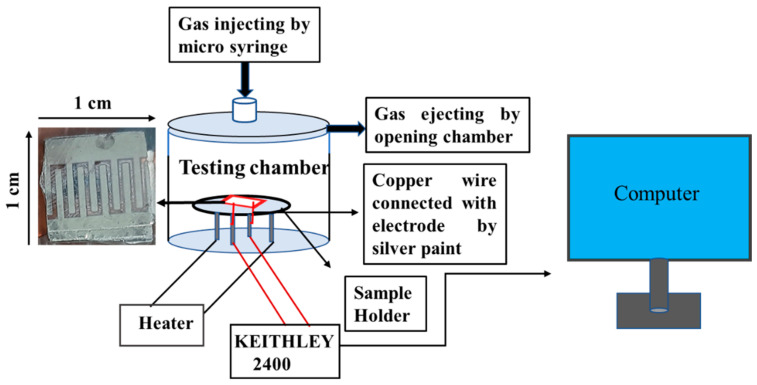
Schematic diagram of gas monitoring set up with the fabricated device.

**Figure 7 micromachines-14-00728-f007:**
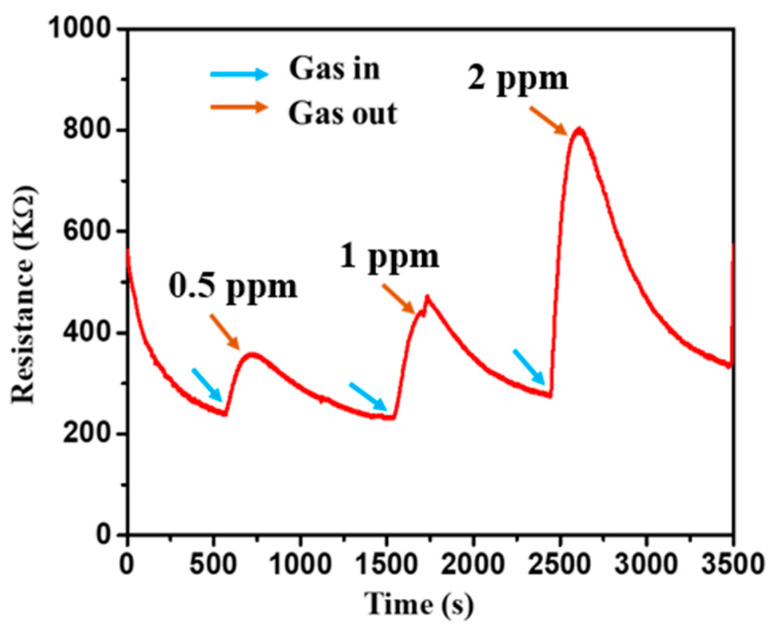
The sensor response of the SnO_2_ device at room temperature at different concentrations of 0.5, 1, and 2 ppm of NO_2_ at room temperature.

**Figure 8 micromachines-14-00728-f008:**
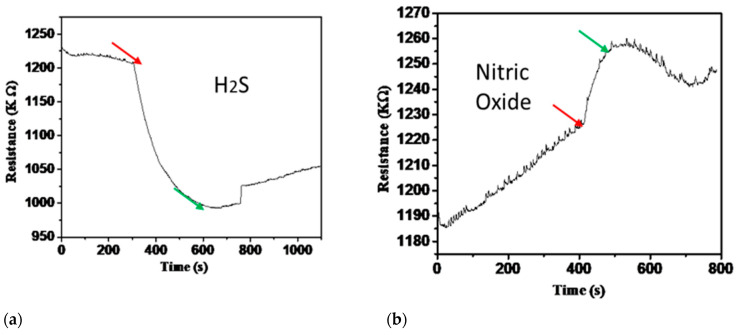
(**a**–**d**) The sensor performance of 1 ppm of H_2_S, NO, NO_2_, and NH_3_ at 150 °C. (**e**) -Sensor response for one ppm of NO_2_ at 50 °C.

**Table 1 micromachines-14-00728-t001:** Parameters of analysed XRD spectra.

Lattice Parameters (Å)	a = 4.739, b = 4.739, c = 3.1879
Angle	α = 90°, β = 90°, γ = 90°
χ ^2^	0.083
Volume (Å)^3^	71.959

**Table 2 micromachines-14-00728-t002:** Comparison table for different gases and materials.

S. No.	Materials	Experimental Process	Measuring Temperature (°C/Gas (ppm)	Sensing Parameter			References
				Sensitivity (S = Rg − Ra/Ra) × 100	Response Time (s)	Recovery Time (s)	
1	SnO_2_	Sol-gel, spin-coating, annealing,	RT/NO_2_ (0.5)	50	84	320	This work
2	SnO_2_	Sol-gel, spin-coating, annealing	RT/NO_2_ (1)	102	124	284	This work
3	SnO_2_	Sol-gel, spin-coating, annealing	RT/NO_2_ (2)	190	184	432	This work
4	SnO_2_	Sol-gel, spin-coating, annealing	150 °C/H_2_S (1)	21	234	No	This work
5	SnO_2_	Sol-gel, spin-coating, annealing	150 °C/NO (1)	2	64	no	This work
6	SnO_2_	Sol-gel, spin-coating, annealing	150 °C/NO_2_ (1)	17	60	n	This work
7	SnO_2_	Sol-gel, spin-coating, annealing	50 °C/NO_2_ (1)	53	74	No	This work
8	SnO_2_	Sol-gel, spin-coating, annealing	150 °C/NH_3_(1)	16	50	no	This work
9	GO/SnO_2_ nanocomposites	Electro-spinning and calcination procedure	120 °C/Formaldehyde (100)	-	32	-	[27]
10	Ni doping of SnO_2_ nanoparticles	Hydrothermal method	200 °C/Formaldehyde (100)	-	130	-	[28]
11	SnO_2–_rGO; Pd hydrogel	-	200 °C/NO_2_ (4)	185	8	-	[29]
12	SnO_2–_ZnO	-	200 °C/NO_2_ (400 ppb)	-	300	300	[30]
13	SnO_2_/Graphene	-	150 °C/NO_2_(10 ppb)	-	43	408	[31]
14	SnO_2_/SnS	-	25 °C/NO_2_ (1 ppb)	-	1800	36 (UV)	[32]
15	SnO_2_/SnS	-	25 °C/NO_2_	-	1800	36 (UV)	[33]
16	SnO_2_ film	Sol-gel	100 °C/NO_2_ (500 ppm)	10 (response)	-	-	[34]
17	SnO_2_	Sol-gel	UV radiation/NO_2_ (5 ppm)	360 (response)	276	-	[35]
18	SnO_2_	Thermal evaporation	200 °C/NO_2_ (0.5 ppm)	-	43	18	[36]
19	SnO_2_	Sputtering	100 °C/NO_2_ (50 ppm)	4300 (response)	126	1152	[37]
20	Mesoporous SnO_2_	Inkjet printing	175 °C/NO_2_ (5 ppm)	11,507 (response)	-	-	[38]

## Data Availability

Data will be made available upon reasonable request.

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
