# Peer review of "SnO_2_-Based NO_2_ Gas Sensor with Outstanding Sensing Performance at Room Temperature"

_micromachines, 2023, doi:10.3390/mi14040728_

Round 1
Reviewer 1 Report
1) Some relevant references must be included in the manuscript:
Kaur, J., Roy, S. C., & Bhatnagar, M. C. (2007). Highly sensitive SnO2 thin film NO2 gas sensor operating at low temperature. Sensors and Actuators B: Chemical, 123(2), 1090-1095.
Hyodo, T., Urata, K., Kamada, K., Ueda, T., & Shimizu, Y. (2017). Semiconductor-type SnO2-based NO2 sensors operated at room temperature under UV-light irradiation. Sensors and Actuators B: Chemical, 253, 630-640.
Choi, Y. J., Hwang, I. S., Park, J. G., Choi, K. J., Park, J. H., & Lee, J. H. (2008). Novel fabrication of an SnO2 nanowire gas sensor with high sensitivity. Nanotechnology, 19(9), 095508.
Devabharathi, N., M. Umarji, A., & Dasgupta, S. (2020). Fully inkjet-printed mesoporous SnO2-based ultrasensitive gas sensors for trace amount NO2 detection. ACS Applied Materials & Interfaces, 12(51), 57207-57217.
Sharma, A., Tomar, M., & Gupta, V. (2011). SnO2 thin film sensor with enhanced response for NO2 gas at lower temperatures. Sensors and actuators B: chemical, 156(2), 743-752.
2) The experimental device is missing and must be included and explained.
3) The procedure for the evaluation of the response time, recovery time must be explained.
4) All the experiments must be performed in the chamber with controlled humidity . SnO2 is very sensitive toward humidity.
5) The sensing mechanism must be detailed. The explanation sounds vague.
6) The conclusions must be reinforced.
7) The novelty of this manuscript is not clear.
Author Response
1) Some relevant references must be included in the manuscript:
Reviewer #1: 1) Some relevant references must be included in the manuscript:
Kaur, J., Roy, S. C., & Bhatnagar, M. C. (2007). Highly sensitive SnO2 thin film NO2 gas sensor operating at low temperature. Sensors and Actuators B: Chemical, 123(2), 1090-1095.
Hyodo, T., Urata, K., Kamada, K., Ueda, T., & Shimizu, Y. (2017). Semiconductor-type SnO2-based NO2 sensors operated at room temperature under UV-light irradiation. Sensors and Actuators B: Chemical, 253, 630-640.
Choi, Y. J., Hwang, I. S., Park, J. G., Choi, K. J., Park, J. H., & Lee, J. H. (2008). Novel fabrication of an SnO2 nanowire gas sensor with high sensitivity. Nanotechnology, 19(9), 095508.
Devabharathi, N., M. Umarji, A., & Dasgupta, S. (2020). Fully inkjet-printed mesoporous SnO2-based ultrasensitive gas sensors for trace amount NO2 detection. ACS Applied Materials & Interfaces, 12(51), 57207-57217.
Sharma, A., Tomar, M., & Gupta, V. (2011). SnO2 thin film sensor with enhanced response for NO2 gas at lower temperatures. Sensors and actuators B: chemical, 156(2), 743-752.
Response – Authors are very thankful to the reviewer for their helpful and valuable suggestions. The references are included and the manuscript has been changed thoroughly to address the above-said things.
Reviewer #1: 2) The experimental device is missing and must be included and explained.
Response – We explained the fabrication method of the device with a schematic diagram (figure 6), and the SEM of the device is shown in Figure 3 (c). It has also been explained.
For the synthesis of SnO2 film, a 1×1 cm2 glass wafer was cleaned using acetone, IPA, and DI water using ultrasonication. After that, the dispersed tin oxide in ethanol was spin-coated, and the sample was kept in a box furnace for annealing at nearly 500°C for two hours. Then, the aluminium electrode was thermally deposited using a comb-like mask with a finger gap of 0.5 mm.
Reviewer #1: 3) The procedure for the evaluation of the response time, recovery time must be explained.
Response- We thank the reviewer for the valuable suggestions. The sensor response for an oxidising gas such as NO2 is defined as S = (Rg – Ra)/Ra, where Ra and Rg are the resistances of the sensor in the presence of atmospheric air and target gas, respectively.
The sensor's time to reach 90% of its maximum resistance value while exposed to the target oxidising gas was used to measure the response time. Upon the achievement of the maximum resistance value, the target gas was evacuated from the test chamber, and the sensor was allowed to get back to its starting resistance value in atmospheric air while maintaining at the temperature. In the presence of atmospheric air, the sensor's recovery time is defined as the amount of time it takes to reacquire 10 % of the initial resistance value.
Reviewer #1: 4) All the experiments must be performed in the chamber with controlled humidity. SnO2 is very sensitive toward humidity.
Response- Authors would like to thank the reviewer for their valuable suggestion. We have done all the measurements at room temperature for the gas sensor. The room has a controlled atmosphere.
Reviewer #1: 5) The sensing mechanism must be detailed. The explanation sounds vague.
Response- We thank the reviewer for the valuable suggestions. SnO2 is an n-type semiconductor, and NH3, NO, H2S, and NO2 (oxidising gas) are used in the present work. When the device is exposed to the NO2 gas, the resistance increases. The gas sensing mechanism depends on the adsorption and desorption of NO2 gas (oxidising gas) on the SnO2 (n-type) film surface. NO2 traps the free electrons from the SnO2 surface and increases the sensor resistance. At room temperature, O2- is present on the SnO2 surface, but NO2 gas molecules react directly with Sn ionic sites instead of oxygen species. The reaction is -
NO2 + Sn2+ (Sn3+-NO2) (Sn3+-O-) + NO
The reduction in the concentration of charge carriers from the conduction band of SnO2 upon adsorption results in an increase in sensor resistance. When gas is released due to desorption, the device gains initial condition at room temperature called recovery.
Reviewer #1: 6) The conclusions must be reinforced.
Response- We thank the reviewer for the valuable suggestions. The conclusion has been modified in the revised manuscript.
Reviewer #1: 7) The novelty of this manuscript is not clear.
Response- We thank the reviewer for the valuable suggestions. The novelty of this manuscript is-
- Fabricated device was selectively sensitive to NO2 gas,
- Synthesis of SnO2 is easy and elements are earth-abundant and low-cost,
- Device is easy to fabricate (Sol-gel and spin-coat techniques),
- Fabricated device operates at room temperature.
Reviewer 2 Report
Comments:
From the TEM analysis, the authors have shown that their SnO2 nanoparticles are around 10-15 nm size. In the SEM image, the nanoparticles has nucleated to micron size which the authors has claim due to solvent. Did the authors do any investigation on different nucleation formation of the nanoparticles?
Minor comments:
1) Can the authors take note of the superscript and subscript used through this manuscript. SnO2 to SnO2, m3 to m3 etc.
2) ACs. Air conditioners? AC may also mean alternative current.
3) Line 80-82. Informal sentence structure used.
4) Line 90-91. “In this synthesis method, first, 100 ml distilled water (18Ω) was filled in 500 round bottom flasks,…” Is it a 500ml?
5) Line 179, is the highest intense peak is at 630 cm-1 or 633cm-1? Differs in image and text.
Author Response
Comments:
Reviewer #2 From the TEM analysis, the authors have shown that their SnO2 nanoparticles are around 10-15 nm size. In the SEM image, the nanoparticles has nucleated to micron size which the authors has claim due to solvent. Did the authors do any investigation on different nucleation formation of the nanoparticles?
Response- Authors would like to thank the reviewer for their valuable suggestion. TEM image shows the particle size of SnO2 powder (~10-15 nm) fabricated by sol-gel method, whereas the SEM image shows the film of SnO2 on glass by spin-coating and then the film was annealed below 500 °C which results in an increment of particle size. Ethanol is used as a dispersion medium for SnO2 particles. Upon heating, the particles join together by forming neck-like structure, which helps in electron conduction.
Minor comments:
Reviewer #2: 1) Can the authors take note of the superscript and subscript used through this manuscript. SnO2 to SnO2, m3 to m3 etc.
Response- Authors would like to thank the reviewer for their valuable suggestion. We have made all corrections in the manuscript.
Reviewer #2: 2) ACs. Air conditioners? AC may also mean alternative current.
Response- Authors would like to thank the reviewer for their valuable suggestion. The full form of ACs here is Air Conditioners and have been corrected in the manuscript.
Reviewer #2: 3) Line 80-82. Informal sentence structure used.
Response- We thank the reviewer for the valuable suggestions, and the sentences have been restructured in the manuscript.
Reviewer #2: 4) Line 90-91. “In this synthesis method, 100 ml distilled water (18Ω) was first filled in 500 round bottom flasks…” Is it a 500ml?
Response-We thank the reviewer for the valuable suggestions. It was written by mistake. It is 500 ml and we have corrected it in the revised manuscript.
Reviewer #2: 5) Line 179, is the highest intense peak is at 630 cm-1 or 633 cm-1? Differs in image and text.
Response- We thank the reviewer for the valuable suggestions. The peak is at 633 cm-1, and have corrected it in the revised manuscript.
Round 2
Reviewer 1 Report
The experimental device used for the monitoring of NO2 is not yet described.
How is monitored the level of humidity during the sensing experiments?
Author Response
Reviewer# The experimental device used for the monitoring of NO2 is not described?
Response- We have added sentences (lines 210-218) and figure 6 (lines 231-233)) in the revised manuscript describing these.
As per the reviewer’s comment, we provided all the measuring equipment models used in the measurements and their respective characteristics. A static gas monitoring system of 250 ml stainless steel has been used to test gases made by Ants innovative private limited, India. NO2 and other gases with various concentrations with different ppm were injected through a micro syringe into the test chamber. The formula for measuring the gas concentration was C1V1=C2V2, where C1 and C2 are the cylinder concentration, and V1 and V2 are the volume of the test chamber and micro syringe, respectively. The device’s electrodes were joined by copper wire using high-grade silver paste before the gas sensing measurement was carried out. The resistance value was recorded using KEITHLEY 2400 source meter.
Figure 6. Schematic diagram of gas monitoring set up with the fabricated device
Reviewer# How is monitored the level of humidity during the sensing experiments
Response –We have added lines 216-218 in the revised manuscript.
We have done all the measurements at room temperature of 21°C and humidity of 60% measured by fluke 1620A “Dewk” thermo-hygrometer. We agree with the reviewer’s comment, but unfortunately, we currently do not have the resources to check the humidity effects. However, we appreciate and will consider the reviewer’s suggestions and carry out further studies to include the experimental evidence in the future.
